

# A novel machine learning derived RNA-binding protein gene–based score system predicts prognosis of hepatocellular carcinoma patients

Qiangnu Zhang[1,2,*], Yusen Zhang[3,*], Yusheng Guo[3], Honggui Tang[3], Mingyue Li[1,3] and Liping Liu[1,3]

[1] The Second Clinical Medical College, Jinan University (Shenzhen People's Hospital)
[2] Integrated Chinese and Western Medicine Postdoctoral Research Station, Jinan University, Guangzhou, China
[3] Shenzhen People's Hospital, Shenzhen, China
* These authors contributed equally to this work.

Corresponding authors
Mingyue Li,
li.mingyue@szhospital.com
Liping Liu, liu.liping@szhospital.com

## ABSTRACT

**Background:** Although the expression of RNA-binding protein (RBP) genes in hepatocellular carcinoma (HCC) varies and is associated with tumor progression, there has been no overview study with multiple cohorts and large samples. The HCC-associated RBP genes need to be more accurately identified, and their clinical application value needs to be further explored.

**Methods:** First, we used the robust rank aggregation (RRA) algorithm to extract HCC-associated RBP genes from nine HCC microarray datasets and verified them in The Cancer Genome Atlas Liver Hepatocellular Carcinoma (TCGA-LIHC) cohort and International Cancer Genome Consortium (ICGC) Japanese liver cancer (ICGC-LIRI-JP) cohort. In addition, the copy number variation (CNV), single-nucleotide variant (SNV), and promoter-region methylation data of HCC-associated RBP genes were analyzed. Using the random forest algorithm, we constructed an RBP gene–based prognostic score system (RBP-score). We then evaluated the ability of RBP-score to predict the prognosis of patients. The relationships between RBP-score and other clinical characteristics of patients were analyzed.

**Results:** The RRA algorithm identified 30 RBP mRNAs with consistent expression patterns across the nine HCC microarray datasets. These 30 RBP genes were defined as HCC-associated RBP genes. Their mRNA expression patterns were further verified in the TCGA-LIHC and ICGC-LIRI-JP cohorts. Among these 30 RBP genes, some showed significant copy number gain or loss, while others showed differences in the methylation levels of their promoter regions. Some RBP genes were risk factors or protective factors for the prognosis of patients. We extracted 10 key HCC-associated RBP genes using the random forest algorithm and constructed an RBP-score system. RBP-score effectively predicted the overall survival (OS) and disease-free survival (DFS) of HCC patients and was associated with the tumor, node, metastasis (TNM) stage, α-fetoprotein (AFP), and metastasis risk. The clinical value of RBP-score was validated in datasets from different platforms. Cox analysis suggested that a high RBP-score was an independent risk factor for poor prognosis in

HCC patients. We also successfully established a combined RBP-score+TNM LASSO-Cox model that more accurately predicted the prognosis.

**Conclusion:** The RBP-score system constructed based on HCC-associated RBP genes is a simple and highly effective prognostic evaluation tool. It is suitable for different subgroups of HCC patients and has cross-platform characteristics. Combining RBP-score with the TNM staging system or other clinical parameters can lead to an even greater clinical benefit. In addition, the identified HCC-associated RBP genes may serve as novel targets for HCC treatment.

## INTRODUCTION

Hepatic carcinoma is the malignant tumor with the sixth-highest incidence in the world and is also the fourth-leading cause of cancer death in the world. Approximately 85–95% of primary liver cancers are hepatocellular carcinoma (HCC) (*Qiao et al., 2021*). Due to the insidious onset and inadequate early diagnostic measures, 80% of HCC patients are already at middle or advanced stages when diagnosed, so they have lost their chance at surgery (*Bray et al., 2018*; *Llovet, Burroughs & Bruix, 2003*). The mortality rate of patients with advanced HCC is as high as 80%, the median survival time is less than 1 year, and the 5-year survival rate is less than 20%. Recent years have seen progress in surgical techniques, radiotherapy and chemotherapy techniques, targeted therapeutics, and immunotherapeutic techniques. These advances have brought new hope to patients with middle or advanced HCC, but it is undeniable that the treatment efficacy of advanced HCC is still dismal (*Kanwal & Singal, 2019*).

Many molecular mechanisms underlying the development and progression of HCC remain unclear. Therefore, it is necessary to better understand the molecular mechanisms of HCC and identify key molecules related to HCC, thereby facilitating the early diagnosis of HCC, the search for therapeutic targets, and the individualization of patients' treatment and prognosis determination.

RNA-binding proteins (RBPs) are important proteins involved in posttranscriptional regulatory events. With a variable RNA-binding region and flexible structure, RBPs regulate the metabolic behavior of many RNAs in a dynamic way, including RNA splicing, localization, transport, and stability maintenance (*Lujan, Ochoa & Hartley, 2018*; *Pereira, Billaud & Almeida, 2017*). RBPs can participate in tumorigenesis and tumor progression through posttranscriptional regulation (*Campos-Melo et al., 2014*). Some RBPs are differentially expressed in cancer tissues and paracancerous tissues, and this differential expression is correlated with the prognosis and clinical characteristics of patients (*Yan et al., 2019*). Recent high-throughput data analyses based on The Cancer Genome Atlas (TCGA) suggested that RBPs change in expression in a variety of cancers. Many studies have identified RBPs associated with the occurrence and development of

HCC. For example, *Zhao et al. (2019)* found that the RNA-binding protein RPS3 promotes the proliferation of HCC cells through posttranscriptional regulation of silent information regulator 1 (SIRT1). *Dong et al. (2019)* found that the RNA-binding protein RBM promoted the growth of HCC cells by regulating the production of the circular RNA SCD-circRNA 2. However, most previous studies have focused on the function and mechanism of single RBPs in HCC cells at the *in vitro* level. There has been no systematic overview or clinical-application study on RBPs in HCC. Although some studies investigated the relationship between RBPs and clinical prognosis, these studies were mostly based on a single dataset, so some studies showed inconsistencies and contradictions. We still await consistent RBP-related data that are based on a large cohort and multiple cohorts with clinical application value.

In this study, we integrated and analyzed the expression pattern of RBPs in HCC through multiple large cohorts and acquired a highly consistent and robust HCC-associated RBP expression profiles. Using artificial intelligence (AI)–based methods, we constructed an RBP gene–related molecular prognostic score system (RBP-score) based on HCC-associated RBP genes. RBP-score can effectively evaluate and predict the prognosis of HCC patients. We also used the least absolute shrinkage and selection operator (LASSO)-Cox model to combine the tumor, node, metastasis (TNM) stage and RBP-score to successfully construct a more effective model for predicting the overall survival (OS) of HCC patients. Lastly, this also discusses the genomics and epigenetic changes of some HCC-associated RBP genes. We believe that the results of this study can help predict and evaluate the prognosis of HCC patients, provide molecular insight into the complex mechanism of HCC, and provide potential targets for HCC treatment.

## MATERIALS & METHODS

### HCC cohorts, data collection, and preprocessing

The microarray mRNA expression data (GSE14520, GSE22058, GSE25097, GSE36376, GSE45436, GSE64041, GSE76427, GSE54236, and GSE63898) from the HCC cohorts of nine different centers were obtained from the Gene Expression Omnibus (https://www.ncbi.nlm.nih.gov/geo/). These datasets were selected based on sample size, data quality, and the completeness of probe coverage. The Cancer Genome Atlas Liver Hepatocellular Carcinoma (TCGA-LIHC) data collection including mRNA expression, copy number variation (CNV), and methylation data were downloaded from the NIH GDC Data Portal (https://portal.gdc.cancer.gov/). International Cancer Genome Consortium (ICGC) Japanese liver cancer (ICGC-LIRI-JP) data were obtained from ICGC DCC (https://dcc.icgc.org/releases). The gene identifiers of all the datasets were converted to the latest HUGO gene symbols. The mRNA expression data were subjected to $\log_2$ transformation and normalization. More details about datasets see in Table S1.

### Integrated mRNA expression analysis to identify the HCC-associated RBP genes

To identify the mRNAs with consistent expression patterns, we used the robust rank aggregation (RRA) algorithm (*Kolde et al., 2012*) to integrate the microarray mRNA

expression from all nine centers. The RRA algorithm was implemented within R software (version 3.6.1) with the RobustRankAggreg Package. The HCC RRA list included genes with fold change $> 1.5$ or $< -1.5$ and $P < 0.05$. The genes in the RRA list were consistently upregulated or downregulated in all nine HCC cohorts. By searching the database of RNA binding specificities (RBPDB, http://rbpdb.ccbr.utoronto.ca/) and referring to the study conducted by *Gerstberger, Hafner & Tuschl (2014)*, we obtained a gene list of human RBPs containing 430 genes, and the products of the genes in this list were functionally identified as proteins that can bind RNA. Thus, the genes making up the intersection of the HCC RRA list and the RBP gene list were defined as the HCC-Associated RBP genes. Next, the expression of HCC-associated RBP genes in the nine microarray cohorts was verified using the RNA sequencing data of TCGA-LIHC and ICGC-LIRI-JP. Functional/pathway enrichment analysis and protein–protein interaction analysis were performed on the HCC-associated RBP genes by the Metascape online tools (http://metascape.org/gp/index.html).

## Detecting the genomic and epigenetic changes of HCC-associated RBP genes

TCGA-LIHC thresholded gene-level CNVs were estimated with the GISTIC2 method. The GISTIC2 method was applied using the TCGA FIREHOSE pipeline to produce gene-level copy number estimates. Genes were mapped onto human genome coordinates using UCSC xena HUGO probeMap. The copy number gain or loss frequency of HCC-associated RBP genes was calculated in 370 patients. MC3 gene-level nonsilent mutation data were obtained from the UCSC Xena public data hub (https://xena.ucsc.edu/public/). To analyze the methylation status of HCC-associated RBP gene promoters, the data of the TCGA-LIHC HumanMethylation450 platform were downloaded. DNA methylation beta values are continuous variables between 0 and 1 representing the ratio of the intensity of the methylated bead type to the combined locus intensity. Beta values of the HCC-associated RBP genes TSS200 and TSS1500 locus were integrated and compared between the carcinoma tissues and para-carcinoma tissues.

## Selecting important HCC-associated RBP genes for the gene signature using the random forest algorithm

To obtain a signature with a small number of genes, we needed to further screen for key HCC-associated RBP genes. To achieve this goal, an AI-based machine learning method was used. In brief, the TCGA-LIHC dataset was used as the training set. Patients were divided into 5-year-surviving patients and 5-year-nonsurviving patients. Then we constructed a random forest classifier model. The mRNA expression data of HCC-associated RBP genes were entered in to the classifier model (ntree = 500). The random forest model was constructed in R (version 3.6.1) with the randomForest package. After stratified 10-fold cross-validation (CV = 10), the top 10 HCC-associated RBP genes according to importance value (mean Gini value, cutoff = 5.1) made up the RBP gene signature.

## Building and validating an HCC-associated RBP gene–based prognostic score system

To make the HCC-associated RBP gene expression levels applicable in the clinic, the above 10 key HCC-associated RBP genes were used to construct a prognostic score system (RBP-score). RBP-score was calculated using the formula: RBP-score = $\sum($Gene_score $*$ Gene_Weight), where Gene_Weight is the Gini coefficient provided by the random forest model. Gene_score was determined by the mRNA level and integrated hazard ratio (HR) of the 10 genes. The integrated HR was obtained from the integrated results of Cox regression analyses (for OS) of the 10 genes in three cohorts, TCGA-LIHC, GSE14520, and ICGC-LIRI-JP. If a gene had integrated HR > 1 and mRNA expression > median expression or integrated HR < 1 and mRNA expression < median expression, then the Gene_score of this gene was 1; otherwise, the Gene_score was 0.

To verify the prognostic evaluation ability of RBP-score, we used the lower quartile, median, and upper quartile of RBP-score as cutoffs to divide patients into four groups (Q1, Q2, Q3 and Q4; RBP-score$_{Q1}$ < RBP-score$_{Q2}$ < RBP-score$_{Q3}$ < RBP-score$_{Q4}$). The OS and DFS of each group were analyzed and compared. HCC patients were divided into subgroups based on age, sex, TNM stage, $\alpha$-fetoprotein (AFP), and other clinical parameters. Patients in each subgroup were divided into the high RBP-score group and low RBP-score group (cutoff = median of RBP-score). The OS and DFS of each group were compared again. *Roessler et al. (2010)* used a metastasis gene signature to group GSE14520 patients into patients with high invasion risk and patients with low metastasis risk. *Chen et al. (2017)* used a six-gene metastasis signature to assess the risk of metastasis among patients in 12 HCC cohorts. The metastasis risk score was calculated using the Gene Set Variation Analysis (GSVA) method (*Hanzelmann, Castelo & Guinney, 2013*). Based on the median value of the metastasis risk score, we divided our patients into the high-metastasis-risk group and the low-metastasis-risk group. RBP-score was compared between the high-metastasis-risk group and the low-metastasis-risk group.

## LASSO-Cox model based on RBP-score and TNM stage

In the TCGA-LIHC dataset, we used the combination of RBP-score and TNM stage to construct a LASSO-Cox model for predicting the 1-year, 3-year, and 5-year OS of HCC patients. The establishment of the LASSO-Cox model and the visualization of the nomogram model were done in R (version 3.6.1) with the hdnom package. Calibration plots and decision curves were simultaneously plotted to verify the effect of the evaluation model.

## Statistics

Statistical analysis was performed using R (version 3.6.1) statistical software.
The differences in normally distributed data between two groups were analyzed using the independent-sample t test, and the differences in nonnormally distributed data were analyzed using the Wilcoxon test. The relationships of each single HCC-associated RBP gene and RBP-score with the OS and DFS of the patients were analyzed by Kaplan-Meier survival analysis with the log-rank test. The HRs of single HCC-associated RBP genes,
RBP-score, and other clinical indicators for OS were obtained by univariate and multivariate Cox analysis. The relationships between RBP-score and clinical features, such as patient AFP and TNM stage, were confirmed by the chi-squared test. $P < 0.05$ was defined as a statistically significant difference.

## RESULTS

### Identification and mRNA expression profiles of a panel of highly consistent HCC-associated RBP genes

The design and workflow of this study are shown in Fig. S1. The mRNA microarray expression data of the nine HCC cohorts were integrated using the RRA algorithm. The list of integrated differentially expressed genes (cancer tissue *vs.* normal tissue, integrated fold change > 1.5 or < −1.5, and adjusted $P < 0.05$) was named the HCC-RRA list. This list contained 1,326 genes with consistent and significantly differential expression (cancer tissue *vs.* normal tissue) across all nine microarray datasets. A list of 430 human RNA-binding proteins (RBP list) was obtained from the database of RNA binding specificities (RBPDB, http://rbpdb.ccbr.utoronto.ca/) and *Roessler et al. (2010)*. By taking the intersection of the HCC-RRA list and the RBP list, we identified 30 RBP mRNAs with consistent differential expression in the nine HCC cohorts, which we defined as HCC-associated RBP mRNAs (Fig. 1A). The differential expression (cancer tissue *vs.* normal tissue) of these 30 RBP mRNAs in the nine HCC microarray datasets is shown in Fig. 1B. Among them, eight RBP mRNAs showed low expression in HCC tissues ($P < 0.05$), while other 22 RBP mRNAs showed high expression in HCC tissues ($P < 0.05$). Next, we verified the expression of all 30 RBP mRNAs in two HCC RNA sequencing datasets (TCGA-LIHC and ICGC-LIRI-JP), which results were highly consistent with the integrated microarray results (Figs. 1C and 1D). We also analyzed the expression of all 30 RBP mRNAs in tissues at different TNM stages (Figs. 1E and 1F). The expression of some RBPs (such as XPO5 and CPEB3) showed significant differences between the early stage and the advanced stage, and this difference was consistent with the changing trend between normal tissues and tumor tissues. Therefore, these RBPs are likely to play a role in cancer promotion or tumor suppression. Principal component analysis (PCA) based on TCGA-LIHC and ICGC-LIRI-JP showed that the mRNA expression profiles of all 30 RBP genes in HCC could effectively distinguish tumor tissue from normal tissue (Figs. 1G and 1H). The above evidence strongly suggests that the 30 RBP genes identified in our study are HCC-related and have further research value. We used the Metascape tool to perform routine gene function/pathway enrichment analysis and protein–protein interaction network analysis on these 30 RBP genes. The results are shown in Fig. S2. Briefly, the functions of these 30 genes were enriched in RNA splicing, RNA metabolism, RNA localization, and RNA stability.

### Genetic variation and epigenetics changes of 30 HCC-associated RBP genes in HCC

In addition to the mRNA expression patterns of the 30 HCC-associated RBP genes, we undertook a simple exploration of the genetic variation and epigenetic changes of these

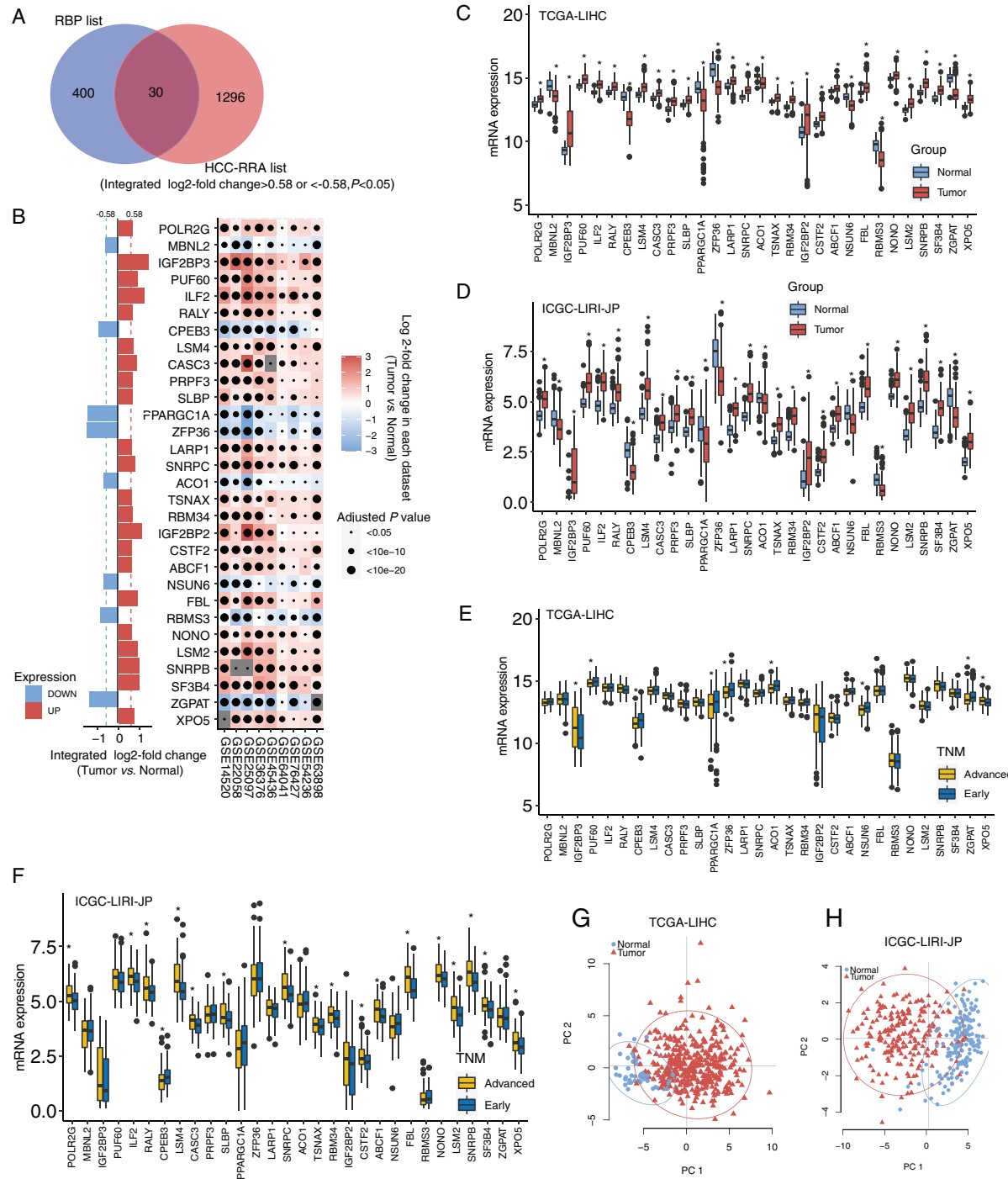

**Figure 1 Identification for HCC-associated RBP genes.** (A) Thirty highly consistent HCC-associated RBP genes were indentified in HCC cohorts by robust rank aggregation algorithm; (B) mRNA expression profiles of thirty highly consistent HCC-associated RBP genes in nine microarray datasets; (C and D) mRNA expression profile of 30 RBP genes were confirmed in Two RNA sequencing datasets; (E and F) the expression of all thirty RBP mRNAs in tissues at different TNM stages; (G and H) Thirty RBP genes in HCC could effectively distinguish tumor tissue from normal tissue through principal component analysis. *$P < 0.05$ compared with Normal tissue or patients with early stage.

30 genes. We first summarized the incidence of CNV (Fig. S3A) and found that the frequency of copy number gain of genes such as PUF60, ILF60, PRPF3, TSNAX, RBM34, and SF3B3 exceeded 50% in the TCGA-LIHC dataset (370 patients), which may partially explain why the mRNAs of these genes were highly expressed in HCC tissues. The mRNA levels of CPEB3, PPARGC1A, and ACO1 were low in HCC tissues. Accordingly, these genes had a high copy number loss frequency. Gene-level nonsilent mutation data from TCGA-LIHC showed the mutation (SNV and indel) frequency of 30 RBP genes among 363 patients. The frequency of nonsilent mutation in the 30 RBP genes was low. The mutation frequencies of the CPEB3, PPARGCA1, and XPO5 genes were relatively high but were only approximately 1% (Fig. S3B). Finally, we used the 450K methylation data of TCGA-LIHC to estimate the methylation levels of the promoter regions of the 30 RBP genes in normal tissues and HCC tissues. The promoter regions of 10 RBP genes showed significant differences in methylation level between HCC tissues and normal tissues, and the methylation levels in the promoter regions of these genes showed a negative correlation with their respective mRNA expression levels (Fig. S3C and Fig. S4). However, except for the relatively high correlation between the methylation level of the promoter region and mRNA expression in ILF2 (r = −0.4, $P < 0.05$), the correlation between the mRNA expression and the methylation of the promoter region was not very strong.

## mRNA expression changes of HCC-associated RBP genes indicates the survival of HCC patients

Next, we explored the clinical value of these 30 HCC-associated RBP genes. We first analyzed the association between the mRNA expression data of each single RBP gene and prognosis in three HCC cohorts (GSE14520, TCGA-LIHC, and ICGC-LIRI-JP). The survival analysis results of 30 RBP genes based on the Cox proportional hazard model are shown in Figs. 2A and S5. The median value of mRNA expression was used as a cutoff. The Kaplan-Meier curve of OS rate and DFS for each HCC-associated RBP gene are shown in Figs. 2B, 2C and Figs. S5, S6 (only the results with log-rank test $P < 0.05$ were shown). The integrated analysis results of the three datasets are shown in Fig. 2D. The results showed that performance of genes such as CSTF2, SF3B4, PPARGCA1, and RALY was consistent between the three datasets. Based on the above evidence, we believe that some genes among the 30 HCC-associated RBP genes are closely related to the survival of HCC patients.

## A molecular prognostic score system based on HCC-associated RBP genes developed using machine learning

To increase the clinical application prospects of the RBP gene expression data, we needed to establish a signature containing as few HCC-associated RBPs as possible. Random forest, a machine learning algorithm, was used to select candidates based on the feature importance of each RBP genes in TCGA-LIHC. In total, 10 HCC-associated RBPs with the highest feature importance for 5-year overall survival were selected by the random forest algorithm. Next, an RBP gene–based molecular prognostic score system (RBP-score)

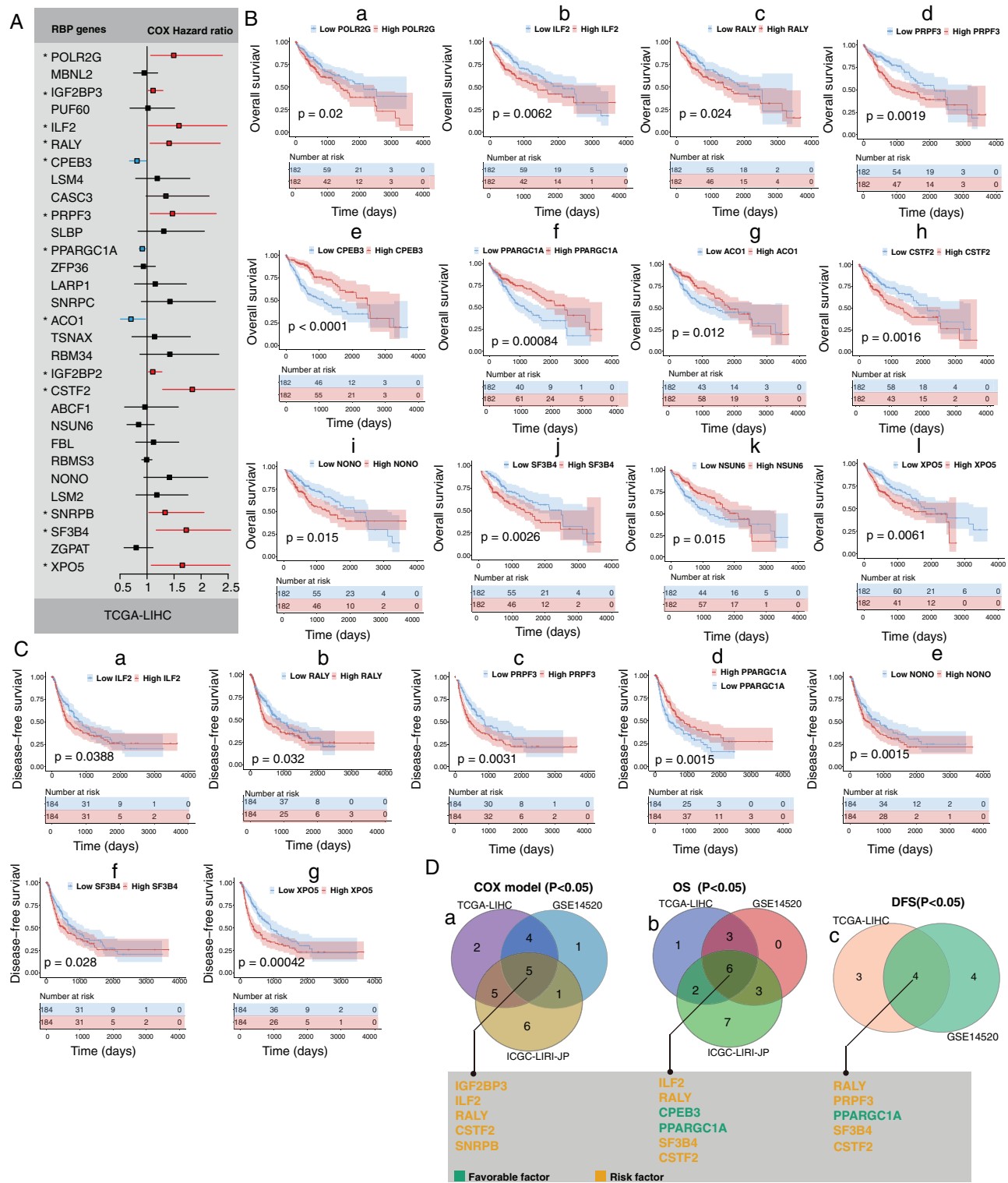

**Figure 2 Part of 30 HCC-associated RBP genes indicated survival of patients with HCC.** (A) Hazard ratio of 30 HCC-associated RBP genes for overall survival in TCGA-LIHC was calculated by Cox model; (B) Kaplan–Meier curves of RBP genes that associated overall survival in TCGA-LIHC; (C) Kaplan–Meier curves of RBP genes that associated disesase-free survival in TCGA-LIHC; (D) Results of integrated survival analysis for 30 HCC-associated RBP genes the three HCC datasets.

**Table 1  The integrated HR value and gene weight of 10 RBPs genes for RBP-score calculation.**

| Gene symbol | HR in GSE14520 | HR in TCGA-LIHC | HR in ICGC-LIRI-JP | Integrated HR value | Gene weight | |
|---|---|---|---|---|---|---|
| PRPF3 | 1.45 | 1.53 | 0.60 | >1 | 9.24 | 10 key RBP genes |
| PPARGC1A | 0.80 | 0.88 | 0.86 | <1 | 6.75 | |
| SLBP | 1.34 | 1.41 | 2.69 | >1 | 6.54 | |
| IGF2BP3 | 1.25 | 1.10 | 1.93 | >1 | 6.17 | |
| SF3B4 | 1.40 | 1.83 | 1.29 | >1 | 6.03 | |
| ILF2 | 1.39 | 1.73 | 1.71 | >1 | 6.03 | |
| CSTF2 | 1.57 | 1.71 | 2.19 | >1 | 5.65 | |
| ACO1 | 0.45 | 0.75 | 0.83 | <1 | 5.23 | |
| CPEB3 | 0.71 | 0.82 | 0.28 | <1 | 5.13 | |
| FBL | 1.49 | 1.12 | 2.16 | >1 | 4.61 | |
| XPO5 | 1.23 | 1.62 | 2.34 | >1 | 4.56 | Others |
| NONO | 1.31 | 1.54 | 3.68 | >1 | 4.53 | |
| RBMS3 | 0.87 | 0.95 | 0.35 | <1 | 4.50 | |
| SNRPB | 1.52 | 1.32 | 2.29 | >1 | 4.41 | |
| IGF2BP2 | 1.13 | 1.09 | 1.24 | >1 | 4.40 | |
| TSNAX | 1.04 | 1.03 | 0.91 | >1 | 4.38 | |
| RBM34 | 1.98 | 1.27 | 1.30 | >1 | 4.32 | |
| LSM2 | 1.11 | 1.20 | 2.65 | >1 | 4.21 | |
| POLR2G | 1.27 | 1.47 | 1.69 | >1 | 4.09 | |
| NSUN6 | 1.06 | 0.87 | 0.57 | <1 | 4.07 | |
| LARP1 | 0.85 | 1.23 | 1.84 | >1 | 4.07 | |
| ZFP36 | 0.87 | 0.96 | 0.74 | <1 | 4.04 | |
| CASC3 | 1.22 | 1.15 | 1.94 | >1 | 3.91 | |
| PUF60 | 1.17 | 1.12 | 1.58 | >1 | 3.86 | |
| MBNL2 | 0.90 | 0.97 | 0.71 | <1 | 3.82 | |
| LSM4 | 0.96 | 1.17 | 1.66 | >1 | 3.71 | |
| ABCF1 | 0.97 | 1.07 | 1.78 | >1 | 3.69 | |
| ZGPAT | 0.86 | 0.92 | 0.87 | <1 | 3.68 | |
| SNRPC | 1.35 | 1.28 | 1.94 | >1 | 3.67 | |
| RALY | 1.43 | 1.34 | 2.56 | >1 | 3.47 | |

was built using the formula: RBP-score = $\sum$(Gene_score $^*$ Gene_Weight). Gene_score was calculated based on the expression level of each of the 10 HCC-associated RBP genes in the samples and the corresponding integrated HR value. Gene_Weight was the Gini coefficient provided by the random forest model. The integrated HR value and Gene_Weight of each of the 10 RBPs are shown in Table 1.

## Validation of the RBP gene–based molecular prognostic score system in an independent cohort

Next, we investigated the association between RBP-score and prognosis in the TCGA-LIHC, GSE14520, and ICGC-LIRI-JP cohorts. The RBP-scores of HCC tissues in
patients were calculated according to the above formula. Patients were divided into four groups using the lower quartile, median, and the upper quartile of RBP-score as the cutoffs (Q1, Q2, Q3 and Q4; RBP-score$_{Q1}$ < RBP-score$_{Q2}$ < RBP-score$_{Q3}$ < RBP-score$_{Q4}$). The Kaplan-Meier curve of the OS rate of patients in each group is shown in Figs. 3A–3C, and a trend of OS decrease with RBP-score increase was clearly observed. Receiver operating characteristic (ROC) analysis of RBP-score for predicting 1-year OS, 3-year OS, and 5-year OS were all >65% in all datasets (Figs. 3D–3F). In TCGA-LIHC and GSE14520, a higher RBP-score also indicated poor DFS (Figs. 3H–3G). The results of the chi-squared test of RBP-score and other clinical features of HCC patients are shown in Tables 2 and 3 (TCGA-LIHC and GSE14520). It was found that patients with higher RBP-scores had higher proportion of AFP > 300 ng/mL, advanced TNM stage (III-IV), CLIP advanced (>3), size > 5 cm, and vascular invasion. Roessler's metastasis signature grouped patients of GSE14520 into the high-metastasis-risk and low-metastasis-risk groups, and we found that patients with high metastasis risk had higher RBP-scores (Fig. 3I, $P < 0.05$). We also evaluated the metastasis risk of more than 2,000 patients in 12 HCC cohorts using Chen's six-gene signature and the GSVA algorithm. Except in the GSE46444 cohort, the RBP-score of patients with high metastasis risk was significantly higher than that of patients with low metastasis risk (Fig. 3J). This means that RBP-score is closely correlated with the metastasis risk of patients. In TCGA-LIHC and GSE14520, Cox proportional hazards analysis was combined with other clinical characteristics of patients (Tables 4 and 5), the results suggest that RBP-score is an independent risk factor for poor OS in HCC patients (HR$_{TCGA-LIHC}$ = 2.57, HR$_{GSE14520}$ = 1.66, $P < 0.05$).

## Subgroup analysis of the prognostic value of the RBP gene–based molecular prognostic score system

Next, we performed subgroup-survival analysis. TCGA-LIHC patients were stratified into subgroups based on clinical characteristics such as age, AFP level, the presence of hepatitis B virus or hepatitis C virus, and TNM stage. Figure 4 shows that with the median as the cutoff, RBP-score also effectively predicted OS in each subgroup. We performed similar subgroup-survival analyses in GSE14520 and ICGC-LIRI-J. Excluding the subgroups with a distribution bias, RBP score predicted OS in most subgroups (Figs. S8 and S9). Even in the same clinical stage, the RBP score effectively predicted OS. Therefore, this RBP gene–based molecular prognostic score system has universal applicability.

## Combined application of the RBP gene–based molecular prognostic scoring system and the TNM clinical staging system

Using ROC curves, we compared RBP-score and TNM stage, another independent risk factor for HCC, for their accuracy in predicting OS in TCGA-LIHC, GSE14520, and ICGC-lIRI-JP (Fig. S10). Based on the accuracy in predicting OS in these three cohorts, RBP-score was not inferior to TNM stage. Using the LASSO-Cox model, RBP-score and TNM stage were combined to construct a nomogram that could predict the probability of 3- and 5-year OS in TCGA-LIHC patients (Fig. 5A). The calibration curve of this model suggested that the model had a certain degree of accuracy (Fig. 5B).

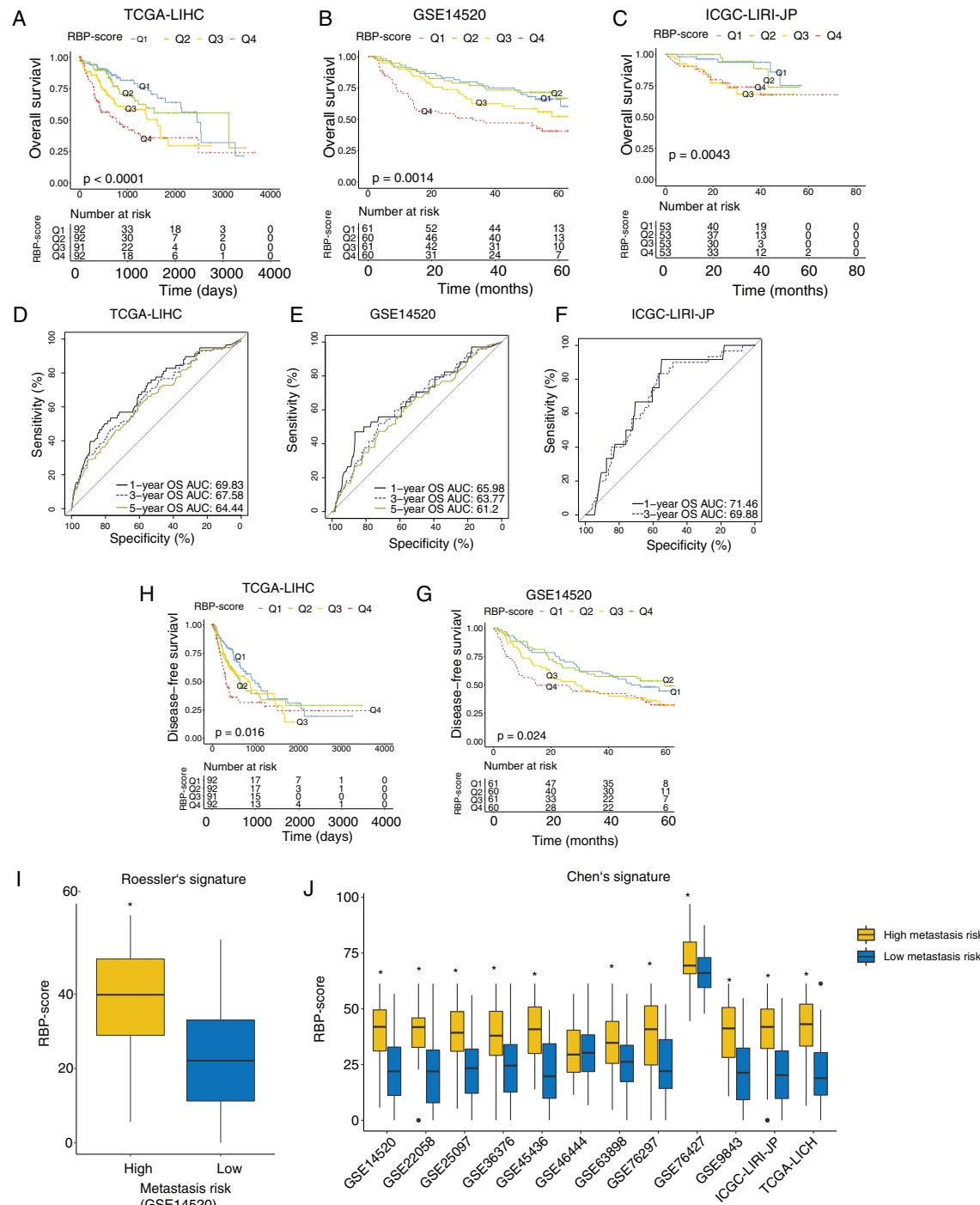

**Figure 3 RBP gene–based molecular prognostic score effectively indicated survival in HCC and assocaited with metastasis.** (A–C) The RBP-scores of HCC tissues in patients were calculated by random forest algorithm and integrated cox model. Patients were divided into four group s using the lower quartile, median, and the upper quartile of RBP-score. Kaplan–Meier curves were created to show overall survival in each group in TCGA-LIHC, GSE14520 and ICGC-LIRI-JP. (D-F) The ROC curves for using RBP-scores to estimate 1-year, 3-year and 5-year overall survival in TCGA-LIHC, GSE14520 and ICGC-LIRI-JP. (H–G) Kaplan–Meier curves were created to show disease-free survival of patients with different RBP-scores in TCGA-LIHC and GSE14520. Metastasis risk of HCC patients were estimated using Roessler's (I) and Chen's signature (J) by gene set variation analysis algorithm. The RBP-score in patients with high metastasis risk and were compared. *P < 0.05 compared with patients with low metastasis risk.

**Table 2 Association of RBP-score with clinical features of HCC patients in TCGA-LIHC.**

| | High RBP-Score | Low RBP-Score | Chi-square | P value |
|---|---|---|---|---|
| **Gender** | | | 0.0496777 | 0.823625 |
| Female | 61 | 58 | | |
| Male | 123 | 126 | | |
| **Age** | | | 1.854706 | 0.1732371 |
| old | 94 | 108 | | |
| Young | 90 | 76 | | |
| **HBV** | | | 0.01340326 | 0.9078329 |
| Negative | 131 | 133 | | |
| Positive | 53 | 51 | | |
| **HCV** | | | 0.02106227 | 0.8846094 |
| Negative | 157 | 155 | | |
| Positive | 27 | 29 | | |
| **AFP** | | | 14.13337 | 0.000170296 |
| >300 ng/mL | 45 | 19 | | |
| ≤300 ng/mL | 91 | 123 | | |
| **Cirrhosis** | | | 0.3972248 | 0.528526 |
| Negative | 56 | 77 | | |
| Positive | 29 | 50 | | |
| **TNM stage** | | | 4.561 | 0.03416726 |
| III-IV | 55 | 36 | | |
| I-II | 118 | 135 | | |
| **Vascular invasion** | | | 9.808369 | 0.007415489 |
| Macro | 12 | 4 | | |
| Micro | 50 | 41 | | |
| None | 86 | 121 | | |

The decision curve (Fig. 5C) suggested that the joint model with RBP-score and TNM stage had a better clinical effect in predicting the OS of HCC patients than either one alone. Based on this evidence, a signature composed of the top 10 HCC-associated RBP genes can effectively indicate the prognosis and clinical characteristics of HCC patients and has clinical application value. The combination of RBP-score and TNM stage has even greater clinical value.

## DISCUSSION

RBPs can affect the expression of many important genes through posttranscriptional regulatory mechanisms to participate in the occurrence and development of tumors. RBP target genes cover a wide range, including cancer-promoting genes, cell cycle/apoptosis regulatory factors, autophagy regulators, and inflammatory factors (*Bish & Vogel, 2014*; *Huang et al., 2018*; *Mohibi, Chen & Zhang, 2019*). Differences in RBPs may be an important reason for the transcriptome heterogeneity of tumor tissues (*Wurth & Gebauer, 2015*). More than 1,500 RBPs have been identified, of which more than 500 are

**Table 3 Association of RBP-score with clinical features of HCC patients in GSE14520.**

|  | High RBP-Score | Low RBP-Score | Chi-square | P value |
|---|---|---|---|---|
| **Gender** | | | | |
| Female | 12 | 19 | 1.331906 | 0.248466 |
| Male | 109 | 102 | | |
| **Age** | | | | |
| old | 19 | 32 | 3.326 | 0.06 |
| young | 100 | 89 | | |
| **multinodular** | | | 1.200202 | 0.273281 |
| Negative | 91 | 99 | | |
| Positive | 30 | 22 | | |
| **AFP** | | | 13.3249 | 0.000262 |
| >300 ng/mL | 70 | 40 | | |
| ≤300 ng/mL | 50 | 78 | | |
| **Cirrhosis** | | | | |
| Negative | 7 | 12 | 0.913854 | 0.339094 |
| Positive | 114 | 109 | | |
| **TNM stage** | | | | |
| III-IV | 32 | 19 | 4.684999 | 0.030427 |
| I-II | 77 | 97 | | |
| **BCLC stage** | | | | |
| B-C | 30 | 23 | 1.445425 | 0.229264 |
| 0-A | 79 | 93 | | |
| **CLIP stage** | | | | |
| >2 | 30 | 18 | 4.137569 | 0.041941 |
| ≤2 | 79 | 98 | | |
| **Tumor szie** | | | | |
| Large | 64 | 44 | 6.346071 | 0.011764 |
| small | 56 | 77 | | |

tumor-related. However, the transcriptome changes in different tumors are specific, so the expression patterns of RBPs in different tumors are also specific. The same RBP may have different roles indifferent tumors (*Wang et al., 2018*). Some studies have attempted to obtain an HCC-related RBP list. For example, *Zhao et al. (2019)* identified 42 HCC-related RBPs using mRNA expression data and clinical data from two HCC cohorts. However, most relevant studies have included small cohorts of HCC patients. The expression pattern of RBPs in different HCC cohorts and different platforms (microarray and RNA-sequencing) may be different. Therefore, it is necessary to integrate multiple HCC cohorts from different platforms to obtain more consistent and robust HCC-associated RBP data. In addition, it is necessary to investigate the clinical application value of RBPs in detail. The present study integrated mRNA expression data of multiple large and cross-platform HCC cohorts. Thirty RBP genes with consistent differential mRNA expression (HCC *vs.* normal tissues) were identified. Then, using a machine learning

**Table 4 Cox proportional hazards analysis based on RBP-score and other clinical fetures for HCC patients in TCGA-LIHC.**

**Univariate**

| Factor | coef | HR | CI | P value |
|---|---|---|---|---|
| gender | −0.23166 | 0.793217 | [0.556645057671346–1.13033227343072] | 0.199847 |
| HBV | −1.03232 | 0.356178 | [0.220752761136395–0.574683899720709] | 2.34E−05 |
| HCV | 0.109094 | 1.115268 | [0.683684948723123–1.81929108773718] | 0.66215 |
| AFP | 0.034394 | 1.034993 | [0.633096172219634–1.69201795651715] | 0.890914 |
| Cirrhosis | −0.23183 | 0.793083 | [0.461914602041643–1.36168004341087] | 0.400582 |
| TNM | 0.918081 | 2.504479 | [1.72749560956676–3.6309307142769] | 1.27E−06 |
| Age | 0.142871 | 1.153581 | [0.81389442988955–1.63503838230947] | 0.422076 |
| vascular_invasion | 0.31683 | 1.372769 | [0.991117855592325–1.90138338368265] | 0.056613 |
| RBP-score | 0.799778 | 2.225046 | [1.55957494274252–3.17447423535556] | 1.03E−05 |

**Multivariate**

| Factor | coef | HR | CI | P value |
|---|---|---|---|---|
| HBV | −1.03852 | 0.353979 | [0.213274032022555–0.58751132651488] | 5.88E−05 |
| TNM | 0.731195 | 2.077561 | [1.42578359477838–3.02728897942979] | 0.000141 |
| RBP-score | 0.947506 | 2.579269 | [1.7484573228048–3.80485564187874] | 1.78E−06 |

**Table 5 Cox proportional hazards analysis based on RBP-score and other clinical fetures for HCC patients in GSE14520.**

**Univariate**

| Factor | coef | HR | CI | P value |
|---|---|---|---|---|
| Gender | 0.620526 | 1.859906 | [0.901677786640907–3.8364584275836] | 0.092998 |
| Multinodular | 0.503508 | 1.654516 | [1.0647680695519–2.57090947822796] | 0.025154 |
| Cirrhosis | 1.628414 | 5.095788 | [1.25552580897008–20.6822170194025] | 0.022706 |
| TNM | 1.291697 | 3.638955 | [2.34027452083652–5.65830749814385] | 9.74E−09 |
| BCLC | 1.306494 | 3.693201 | [2.38162476728294–5.7270715523281] | 5.32E−09 |
| CLIP | 1.16891 | 3.218482 | [2.06883686964941–5.00698120858211] | 2.17E−07 |
| AFP | 0.523637 | 1.688156 | [1.12666496484618–2.52947335377243] | 0.011148 |
| Age | −0.16621 | 0.846868 | [0.506797415733995–1.41513111893331] | 0.525762 |
| Size | 0.731166 | 2.077502 | [1.38609781262485–3.11378936776477] | 0.000398 |
| RBP-score | 0.6607 | 1.936148 | [1.28502186227205–2.9172021000271] | 0.001583 |

**Multivariate**

| Factor | coef | HR | CI | P value |
|---|---|---|---|---|
| TNM | 0.449881 | 1.568126 | [0.761502363827402–3.22916710551] | 0.222207 |
| BCLC | 0.941859 | 2.564745 | [1.08206347399257–6.0790476030062] | 0.032429 |
| CLIP | 0.912235 | 2.48988 | [1.11441424099237–5.56301585083656] | 0.026143 |
| RBP-score | 0.50888 | 1.663428 | [1.06272648285978–2.60367198647845] | 0.026008 |

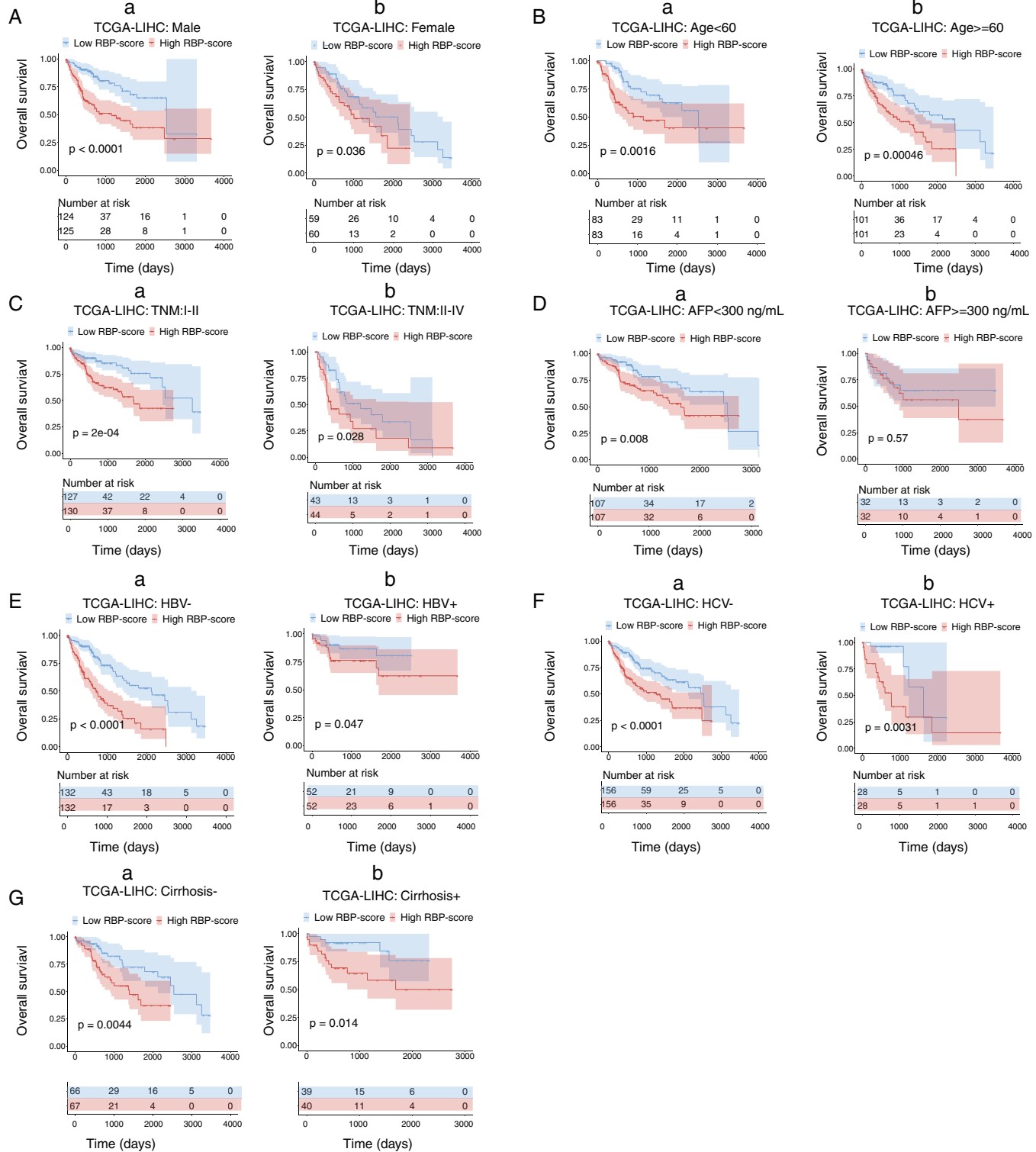

**Figure 4 Kaplan–Meier survival subgroup analysis of patients with HCC according to the RBP-score stratified by clinical characteristics.** In TCGA-LIHC patients were stratified by gender (A), age (B), TNM staging (C), AFP level (D), HBV infection (E), HCV infection (F) and cirrhosis condition (G). The overall survival of patients with different RBP-score in each subgroup were compared.

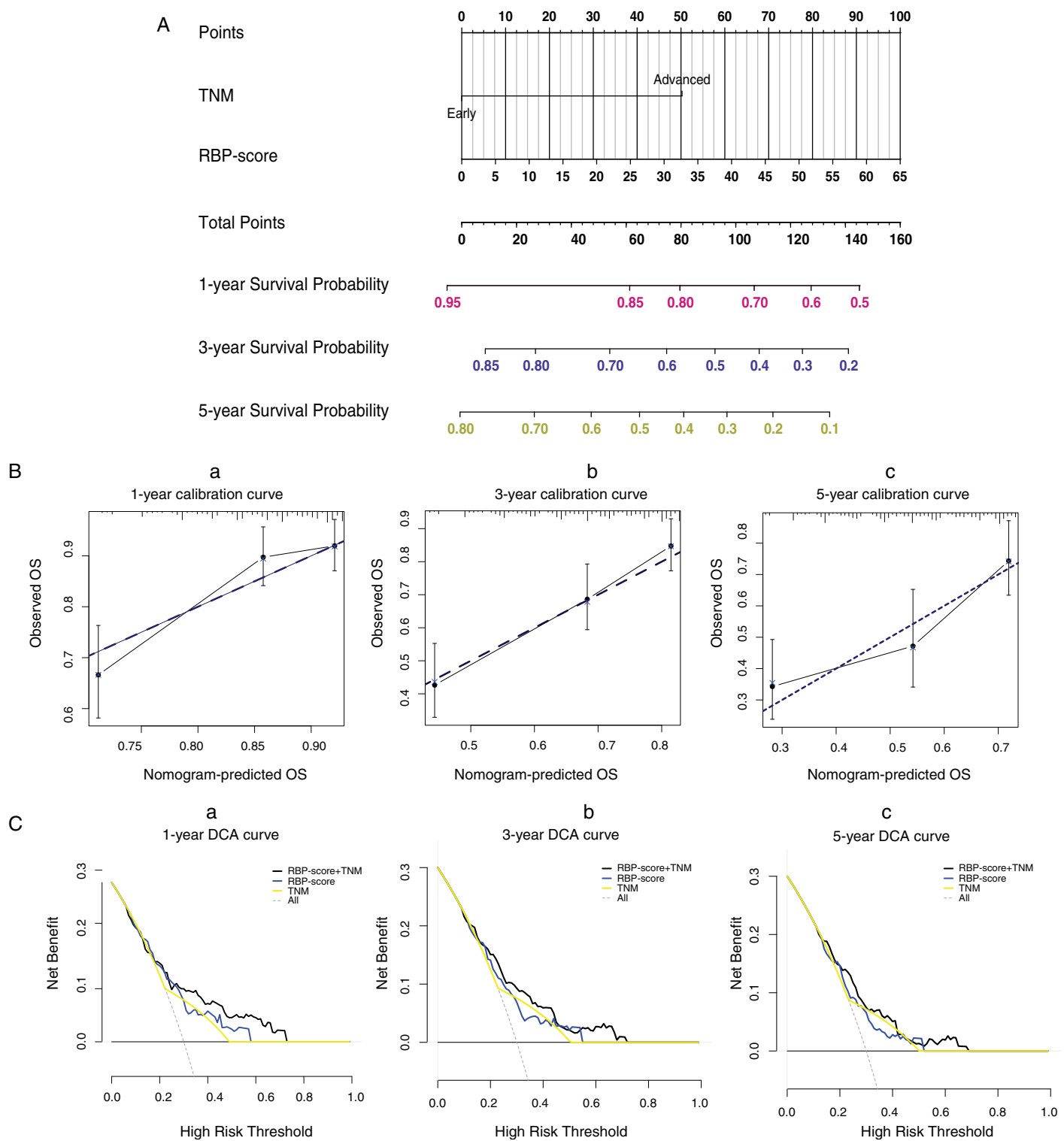

**Figure 5 LASSO-Cox model based on RBP-score and TNM staging was built to predict the probability of 1-year, 3-year and 5 year overall survival in HCC patients.** (A) The LASSO-Cox model based on RBP-score and TNM staging for overall survival prediction was visualized using nomogram; (B) The accuracy of the LASSO-Cox model was indicated by calibration curve. (C) Clinical utility potential of RBP-score/TNM staging combined LASSO-Cox model was shown by decision curve.

technique, the random forest algorithm, we constructed an excellent molecular prognostic score system based on HCC-associated RBPs. This score was closely correlated with the survival and some important clinical features of HCC patients, showing high clinical application potential.

Although RNA-seq is technically more advanced than microarray, relatively few high-quality RNA-seq-based datasets are currently available online excepting TCGA and ICGC. In contrast, high-quality microarray datasets are plentiful and easy to access. Hence we started our study with microarray datasets instead of the RNA-seq datasets. There are many inconsistent results in microarray datasets between the HCC cohorts of different centers and platforms. The expression patterns of the same gene in different microarray datasets inevitably have varying degrees of difference and even show completely opposite results. Therefore, if few cohorts are included in a study, the obtained results are often limited, are not universally applicable, and do not reflect the real-world situation. To solve this problem, we first included the microarray datasets of nine HCC cohorts. Next, we identified 30 RBPs with consistent expression patterns in all nine HCC cohorts among 430 RBPs with well-defined functions using the RRA algorithm and defined them as HC-associated RBPs. The RRA algorithm can effectively integrate multiple microarray data and obtain very robust integration results even if the platforms are different (*Altmae et al., 2017*). Through the RRA algorithm, we not only identified more HCC-specific RBPs but also achieved data dimensionality reduction: the study scope was reduced from 430 RBPs to 30 RBPs, greatly reducing the research burden. Next, the integration results were validated in two RNA sequencing platforms, namely, the TCGA-LIHC and ICGC-LIRI-JP cohorts. The expression patterns of these 30 HCC-associated RBPs were completely consistent with those from the microarrays in the RNA sequencing data. In TCGA-LIHC and ICGC-LIRI-JP, expression of some highly expressed HCC-associated RBP genes in cancer tissues increased in advanced-TNM-stage tissues, while low-expressed HCC-associated RBP genes in cancer tissues decreased in advanced-TNM-stage tissues. These results suggest that these 30 HCC-associated RBPs might be related to the prognosis of HCC patients. PCA analysis showed that the mRNA expression of 30 HCC-associated RBPs effectively distinguished between cancer tissues and normal tissues. Based on the above evidence, we believe that the 30 RBPs identified in this study have HCC specificity.

Before investigating the clinical value of these 30 HCC-associated RBPs, we briefly analyzed the CNV, somatic cell mutation status, and methylation status of the promoter region of each of the 30 RBP genes to provide some explanation for their mRNA expression differences. Some of the 30 genes had higher frequencies of CNV. For example, the copy number gain ratios of ILF2, PRPF3, TSNAX, and RBM34 were all greater than 60%, while the mRNAs of these genes were highly expressed in HCC tissues. Other genes with high mRNA expression in HCC tissues also had different degrees of copy number gain, while several genes with low mRNA expression in HCC tissues, such as CPEB3 and ACO1, had a certain proportion of copy number gain. Therefore, changes in HCC-associated RBP gene expression in HCC may be caused by genomic-level changes. We did not find significant somatic cell mutations in the 30 HCC-associated RBP genes.
Therefore, protein function changes induced by somatic cell mutations seem to have little relation to the roles of these 30 HCC-associated RBPs in HCC. The methylation data suggested that the methylation levels of the promoter regions of the 10 HCC-associated RBPs were different between cancer tissue and para-carcinoma normal tissue.

For example, the IFA-2 promoter methylation decreased in cancer tissues, and the methylation level showed a negative correlation with the mRNA level. This suggests that high ILF-2 mRNA expression may be associated with hypomethylation. Although the other nine genes also had methylation differences in the promoter region, and the methylation level showed a negative correlation with the mRNA level, the correlation was not strong. Thus, the methylation changes in the promoter region may be only a small part of the reason for the expression changes of these RBPs. Of course, the CNV, SNV, and methylation data were only acquired from TCGA-LIHC. If there are other high-quality data available in the future, we believed that a more accurate explanation will be obtained.

The most important objective of this study was to explore the clinical value of these HCC-associated RBP genes. First, we analyzed the relationship between single HCC-associated RBP genes and patient survival and found that the majority of the 30 RBP genes were correlated with the survival of the HCC patients, so each gene's expression may be a risk or protective factor for the survival of patients. These results suggest that these RBPs may play a role in tumor promotion or tumor suppression. In particular, after the integration of TCGA-LIHC, GSE14520, and ICGC-LIRI-JP datasets, we found that RBP genes, including CSPP2, RALY, SF3B4, CPEB3, and PPARGC1A, yielded relatively consistent results in the survival analysis, and their relationship with survival was more convincing. Thus, these genes may be worthy of further investigation.

The clinical value of a single HCC-associated RBP gene is limited. However, it is not practical or economical to measure all 30 HCC-associated RBP mRNAs in clinical practice. Therefore, it was necessary to screen for the most important HCC-associated RBP genes to form a signature that was convenient for clinical use. This study used a random forest algorithm to calculate the importance of each HCC-associated RBP gene in determining the 5-year survival of patients. The random forest algorithm is a commonly used supervising learning technology for screening characteristic genes. It is fast and can also achieve good prediction results without much hyperparameter adjustment.

The modeling process is relatively simple. Therefore, it is suitable for clinical application (*Johann et al., 2019*; *Toth et al., 2019*). Using the random forest algorithm, we calculated the importance of all 30 HCC-associated RBP genes in predicting the 5-year survival of patients. Based on importance value, the 10 most important HCC-associated RBP genes were selected to construct the signature. Some of the top 10 HCC-associated RBP genes have been functionally studied in HCC cells *in vitro*. For example, *Jeng et al. (2008)* confirmed that IGF2BP3 promotes tumor invasion and predicts early recurrence and poor prognosis in hepatocellular carcinoma. *Liu et al. (2018)* reported the role of SF3B4 in promoting the proliferation and invasion of HCC cells. Miao confirmed the inhibitory effect of CPEB3 on HCC cell proliferation (*Miao et al., 2020*). *Du et al. (2019)* pointed out that ILF2 stimulates the malignant phenotype of HCC by stabilizing

CREB. These findings indicate that the 10 HCC-associated RBP genes screened in this study indeed play important roles in the occurrence and development of HCC, so they are worth further exploration in terms of their clinical value.

Based on the Cox survival analysis and random forest model results, a molecular prognostic score system (RBP-score) was constructed based on the mRNA expression values of these 10 HCC-associated RBP genes. RBP-score associates the importance of the 10 HCC-associated RBP genes with the prognosis and clinical characteristics of HCC patients. Patients with different RBP-scores in three different HCC datasets had significant differences in OS and DFS: Patients with higher RBP-scores had poorer OS and DFS. ROC curve analysis showed that the accuracy of RBP-score in predicting OS in patients reached more than 65%, suggesting it has practical clinical value. In the subgroup based on the clinical characteristics of patients, RBP-score still showed good OS-predictive performance. Especially in patients with the same clinical stage, the clinical efficacy of RBP-score was seen. We also found that RBP-score was related to the clinical characteristics of patients, including TNM stage, AFP, and vascular invasion. These data were validated in the HCC datasets of three different platforms (microarray and RNA sequencing). Using the metastasis signatures of *Roessler et al. (2010)* and *Chen et al. (2017)*, we assessed the metastasis risk of HCC patients. As expected, RBP-score was higher in patients with a higher risk of metastasis, suggesting not only that RBP-score be used to predict the risk of metastasis but also that the related RBPs are involved in the metastasis function of HCC cells.

Based on the above evidence, we believe that RBP-score is an effective, cross-platform, universally applicable tool for evaluating the prognosis of patients. We also found that the ability of RBP-score to predict OS was not weaker than that of TNM stage. The LASSO-Cox model that combined RBP-score and TNM stage was more effective than either alone, suggesting that combining RBP-score with existing clinical indicators may benefit patients more.

Three published papers are similar to our report including Tian's, Huang's, and Wang's studies (*Tian et al., 2020*; *Huang et al., 2020*; *Wang et al., 2020*). Similar to our research, these studies established prognostic models based on hub RBP-related genes using public HCC datasets. Tian's model was constructed based on only two RBPs. Huang's and Wang's used six and seven genes, respectively. We selected ten key genes and had only one common gene with Huang's study (PPARGC1A). Both our and their models provided a risk score system to separate into high- and low-risk groups with different survival. The accuracy of our prognostic signature was comparable compared with other studies. The time-dependent area under the ROC curve values is around 0.65 ~ 0.70. But we have some technological advantages. Technically, we enrolled multi-center datasets with large-size samples to identify HCC-related RBP genes. But previous studies only used single datasets. We also used the RRA algorithm to integrate multiple datasets to obtain consistent data among multiple datasets. Previous studies did not consider the consistency of results between multi-center datasets. We use an artificial intelligence algorithm to build the risk model, which is more robust and accurate than the classical COX model used in previous studies. We validated the model in different cohorts and

performed stratified analyses of patients which indicate the universality of our model. Hence our validation strength is higher than other studies.

This study only discussed the clinical value of 30 HCC-associated RBP genes but ignored mechanistic research, which is a limitation of this study. What are the specific functions of these RBPs in HCC? What are the target RNAs that can bind to these RBPs? These are the questions need to be addressed. In our ongoing research, we are investigating the functions and molecular mechanisms of PPARGC1A in HCC. We hope this study can provide clues and candidate subjects for RBP-related studies.

## CONCLUSIONS

In conclusion, this study identified 30 HCC-associated RBP genes. The expression patterns of these genes in HCC tissues were different. The RBP-score constructed based on these HCC-associated RBP genes can effectively evaluate and predict the prognosis of patients. RBP-score was also correlated with some clinical features of patients. The prognostic performance of RBP-score has the characteristic of cross-platform applicability. The combination of RBP-score and other existing clinical indicators can increase its clinical application potential. In addition, given their importance, these HCC-associated RBP genes may become novel targets for HCC treatment or diagnosis.

### Data Sharing Statement

The datasets supporting the conclusions of this article are available in the TCGA data portal (https://portal.gdc.cancer.gov/), the Gene Expression Omnibus (GEO, https://www.ncbi.nlm.nih.gov/geo/), The data portal of Chinese Human Proteome Project (CNHPP) Consortium (http://liver.cnhpp.ncpsb.org/) and ICGC Data Portal (https://icgc.org/).

### Funding

This study was supported by the Science and Technology Innovation Commission of Shenzhen (KQJSCX20180321164801762) and the Shenzhen Science and the Technology Project (JCYJ20180305164841126). The funders had no role in study design, data collection and analysis, decision to publish, or preparation of the manuscript.

### Grant Disclosures

The following grant information was disclosed by the authors:
Science and Technology Innovation Commission of Shenzhen: KQJSCX20180321164801762.
Shenzhen Science and Technology Project: JCYJ20180305164841126.

### Competing Interests

The authors declare that they have no competing interests.

## Author Contributions

- Qiangnu Zhang conceived and designed the experiments, performed the experiments, analyzed the data, prepared figures and/or tables, authored or reviewed drafts of the paper, and approved the final draft.
- Yusen Zhang conceived and designed the experiments, performed the experiments, analyzed the data, prepared figures and/or tables, authored or reviewed drafts of the paper, and approved the final draft.
- Yusheng Guo performed the experiments, analyzed the data, prepared figures and/or tables, and approved the final draft.
- Honggui Tang performed the experiments, analyzed the data, prepared figures and/or tables, and approved the final draft.
- Mingyue Li conceived and designed the experiments, authored or reviewed drafts of the paper, and approved the final draft.
- Liping Liu conceived and designed the experiments, authored or reviewed drafts of the paper, and approved the final draft.

## Data Availability

The microarray data are available at NCBI GEO: (GSE14520, GSE22058, GSE25097, GSE36376, GSE45436, GSE64041, GSE76427, GSE54236, and GSE63898).

The Cancer Genome Atlas Liver Hepatocellular Carcinoma (TCGA-LIHC) data collection including mRNA expression, copy number variation (CNV), and methylation data are available at the NIH GDC Data Portal (https://portal.gdc.cancer.gov/).

International Cancer Genome Consortium (ICGC) Japanese liver cancer (ICGC-LIRI-JP) data were obtained from ICGC DCC (https://dcc.icgc.org/releases).

## Supplemental Information

Supplemental information for this article can be found online at http://dx.doi.org/10.7717/peerj.12572#supplemental-information.

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
