# Peer review of "A novel machine learning derived RNA-binding protein gene–based score system predicts prognosis of hepatocellular carcinoma patients"

_PeerJ, doi:10.7717/peerj.12572_

## Round 0.1 · original submission · Major Revisions

The authors are requested to appropriately cite previously published similar studies and highlight key differences in your current study. Please address comments from the second reviewer related to your experimental design and findings.

·

Basic reporting

Well written and clearly organized!

Experimental design

The research article falls within the scope of the journal and elucidating the role of RNA binding proteins (RBP's) in Hepatocellular carcinoma (HCC), a highly malignant aggressive cancer is critical to guide clinical therapy. The basis for selecting the specific microarray datasets instead of the RNA-seq data is not clearly explained. Please elaborate more on the study is based on those datasets. Also, a recent publication on the role of RBP's in HCC using the RNA-seq data from TCGA and the ICGC HCC cohort identified prognosis-related RBPs. And the majority of the pipeline in this study is similar to the published paper on " A novel RNA binding protein-related prognostic signature for hepatocellular carcinoma". Please cite this paper and also add details on how your study is different, and compare the list of RBP's identified from yours to their findings.

Validity of the findings

The provided data is statistically robust, and the findings are linked to the original research question but most parts of the study seem similar to a recently published analysis. Please add on how your study is different from other similar findings.

Reviewer 2 ·

Basic reporting

The manuscript entitled ‘A novel machine learning derived RNA-binding protein gene–based score system predicts prognosis of hepatocellular carcinoma patients’ by Zhang et al, is an analysis of the potential use of RNA-binding proteins (RBP) for prognosis of HCC. To this end, the authors used multiple publicly available datasets and extracted HCC-associated RBP genes. Using machine learning algorithms, they created an RBP-score system and found that it was predictive of metastasis, predictive of prognosis among other clinical characteristics. Overall, the manuscript is well written, and findings look robust.

Comments
1. Authors should cite literature appropriately. E.g Previous work from Wang et al, BMC cancer, 2020 has performed analysis of RBP in HCC. Authors should check the literature and discuss the similarities / differences of the results. Also, Tian et al, Front. Oncol, 2021.
2. The analysis of copy number variation, somatic mutations, promote region methylation is very crude. Ideally, the reader would like to see some validation. The authors have not clearly pointed out the take home message for this analysis – Figure 2. Besides, in the line 387, authors themselves say that it is not clear whether these contribute any direct changes to expression. Figure 2 may fit supplementary, as it steers the reader from the main message of the story. Thus, authors may think to rearrange the figures.
3. The authors used multiple datasets. Can they briefly describe in methods / present table to show the type of patients e.g. age or the cancer stage of the patents for the reader to grasp if these datasets were of similar group of patients or if there was a large variability?
4. Several figures have grammatical errors e.g. in Figure S1.
5. Line 269, the authors describe the RBP-score, but where are the scores of the gene list provided?
6. Several references seem missing, e.g. for Roessler metastasis signature, Chen’s six gene signature.
7. Line 272, Can the authors present the absolute HR values of the all the genes and not just the 10? They may make two categories.
8. Line 350, nine HCC chosen datasets – what was so special about this choice?

Experimental design

See above

Validity of the findings

See above

Additional comments

See above

---

## Round 0.2 · Minor Revisions

When assessing your paper, the reviewer identified some minor issues.

1- The English language should be improved to ensure that an international audience can clearly understand your text.
2- Please discuss how the current study compares to previously published studies.

Reviewer 2 ·

Basic reporting

The revised version of the manuscript shows some improvement. Not all errors have been corrected. The Figure S1 has spelling error of microarray. Not corrected.

The new paragraph starting at line 438 has grammatical errors. Moreover, authors have only done technical comparison with previous studies. They have not compared what findings/results were similar / different. This latter part is important for readers to understand what makes the current study an advance.

Experimental design

see above

Validity of the findings

see above

Additional comments

see above

---

## Round 0.3 · accepted · Accept

Thank you for your submission.